# Generation of Inducible *BCL11B* Knockout in *TAL1/LMO1* Transgenic Mouse T Cell Leukemia/Lymphoma Model

**DOI:** 10.3390/ijms23094932

**Published:** 2022-04-29

**Authors:** Grzegorz K. Przybylski, Dorota Korsak, Katarzyna Iżykowska, Karina Nowicka, Tomasz Zalewski, Małgorzata Tubacka, Maria Mosor, Danuta Januszkiewicz-Lewandowska, Magdalena Frydrychowicz, Maciej Boruczkowski, Grzegorz Dworacki, Jens van den Brandt, Piotr Grabarczyk, Christian A. Schmidt, Chengwu Zeng, Yangqiu Li

**Affiliations:** 1Institute of Human Genetics, Polish Academy of Sciences, 60-479 Poznan, Poland; dorota.korsak@igcz.poznan.pl (D.K.); katarzyna.izykowska@igcz.poznan.pl (K.I.); karina.nowicka@igcz.poznan.pl (K.N.); mosor@fnp.org.pl (M.M.); 1962dj@gmail.com (D.J.-L.); 2NanoBioMedical Centre, Adam Mickiewicz University, 61-614 Poznan, Poland; tomekz@amu.edu.pl; 3Department of Human Nutrition and Dietetics, Poznan University of Life Sciences, 60-637 Poznan, Poland; malgorzata.tubacka@mail.up.poznan.pl; 4Department of Pediatric Oncology, Hematology and Bone Marrow Transplantation, Karol Marcinkowski University of Medical Sciences, 60-572 Poznan, Poland; 5Department of Immunology, Karol Marcinkowski University of Medical Sciences, 60-806 Poznan, Poland; frydrychowicz@ump.edu.pl (M.F.); maciej.boruczkowski@gmail.com (M.B.); gdwrck@ump.edu.pl (G.D.); 6Central Core & Research Facility of Laboratory Animals, University of Greifswald, 17489 Greifswald, Germany; brandtj@uni-greifswald.de; 7Clinic for Internal Medicine C, University Greifswald, 17489 Greifswald, Germany; pgrabarczyk@yahoo.com (P.G.); christian.schmidt@med.uni-greifswald.de (C.A.S.); 8Key Laboratory for Regenerative Medicine of Ministry of Education, Institute of Hematology, School of Medicine, Jinan University, Guangzhou 510632, China; bio-zcw@163.com (C.Z.); yangqiuli@hotmail.com (Y.L.)

**Keywords:** BCL11B, TAL1, LMO1, CRE-ER/lox, T-ALL, TCL, mouse model

## Abstract

The B-cell CLL/lymphoma 11B gene *(BCL11B*) plays a crucial role in T-cell development, but its role in T-cell malignancies is still unclear. To study its role in the development of T-cell neoplasms, we generated an inducible *BCL11B* knockout in a murine T cell leukemia/lymphoma model. Mice, bearing human oncogenes TAL BHLH Transcription Factor 1 (*TAL1*; *SCL*) or LIM Domain Only 1 (*LMO1)*, responsible for T-cell acute lymphoblastic leukemia (T-ALL) development, were crossed with *BCL11B* floxed and with CRE-ER/lox mice. The mice with a single oncogene *BCL11B^flox/flox^CRE*^tg/tg^*TAL1*^tg^ or *BCL11B^flox/flox^CRE*^tg/tg^*LMO1*^tg^ were healthy, bred normally, and were used to maintain the mice in culture. When crossed with each other, >90% of the double transgenic mice *BCL11B^flox/flox^CRE*^tg/tg^*TAL1*^tg^*LMO1*^tg^, within 3 to 6 months after birth, spontaneously developed T-cell leukemia/lymphoma. Upon administration of synthetic estrogen (tamoxifen), which binds to the estrogen receptor and activates the Cre recombinase, the *BCL11B* gene was knocked out by excision of its fourth exon from the genome. The mouse model of inducible *BCL11B* knockout we generated can be used to study the role of this gene in cancer development and the potential therapeutic effect of *BCL11B* inhibition in T-cell leukemia and lymphoma.

## 1. Introduction

T-cell neoplasms, including T-cell acute lymphoblastic leukemia (T-ALL) and T-cell lymphoma (TCL), divided into peripheral T-cell lymphomas (PTCL) and cutaneous T-cell lymphomas (CTCL), comprise a heterogeneous group of hematopoietic tumors. T-ALL is an aggressive malignancy of early T-cell progenitors characterized by high numbers of blast cells in the bone marrow and peripheral blood, enlargement of mediastinal lymph nodes, and often central nervous system involvement. T-ALL has an incidence of 0.3/100,000 and accounts for approximately 15% of pediatric and 25% of adult ALL cases [1]. PTCL and CTCL originate from mature T-cells, have an incidence of 1.0/100,000 and 0.8/100,000 respectively, and together constitute 15% of all non-Hodgkin lymphomas (NHL) in the USA [1]. Due to the ageing of the population, the incidence of TCL is increasing. These neoplasms often present at an advanced stage at the time of diagnosis, and most commonly have an aggressive clinical course requiring prompt treatment. In contrast to B-cell precursor ALL, which has a very good prognosis with conventional therapy, and B-cell NHL, in which substantial clinical progress has been made with the introduction of monoclonal antibodies such as Rituximab, no comparable advances have been seen in T-ALL and PTCL.

Similar to other types of hematopoietic malignancies, T-cell neoplasms are caused by genetic alterations in hematopoietic precursors (T-ALL) or more mature T-cells (PTCL and CTCL), leading to a variety of changes, including loss of cell cycle control, unlimited self-renewal capacity, impaired differentiation, hyperproliferation and loss of sensitivity to death signals [2,3]. To improve the prognosis in T-cell malignancies, new, gene-targeted therapies have to be developed.

The B-cell CLL/lymphoma 11B gene (*BCL11B*) is a key player in T-cell development [4], but its role in T-cell malignancies is still unclear. Our group originally identified a chromosomal rearrangement in T-ALL involving *BCL11B* [5]. We found a high expression of *BCL11B* in T-ALL and showed that the in vitro suppression of the *BCL11B* gene leads to massive apoptosis in malignant, but not in normal T-cells [5,6]. Therefore, *BCL11B* might be an attractive target for the specific therapy of T cell leukemia and lymphomas.

TAL BHLH Transcription Factor 1 (*TAL1*;*SCL*) and LIM Domain Only 1 (*LMO1*) are oncogenes whose activation plays an important role in the development of a fraction of human T-ALL. *TAL-1* is an essential transcription factor in normal and malignant hematopoiesis. It is required for the specification of the blood program during the development, adult hematopoietic stem cell survival and quiescence, and terminal maturation of select blood lineages [7]. Following ectopic expression, caused by a chromosomal translocation t(1;14) or a fusion to the STIL Centriolar Assembly Protein (*STIL*) exogenous promoter, *TAL-1* contributes to oncogenesis in T-ALL. *TCL1* activities are all mediated through the nucleation of a core quaternary protein complex (TCL1:E-protein:LMO1/2:LDB1) and the recruitment of additional regulators in a lineage- and stage-specific context. *LMO1* belongs to a large family of proteins that are required for many developmental processes and are implicated in the onset or the progression of several cancers, including T cell leukemia, breast cancer and neuroblastoma. It contains two protein-interacting LIM domains that operate through nucleating the formation of new transcriptional complexes and/or by disrupting existing transcriptional complexes to modulate gene expression programs. Through these activities, LMO1 has important cellular roles in processes that are relevant to cancer such as self-renewal, cell cycle regulation and metastasis [8]. Transgenic mice bearing both *TCL1* and *LMO1* human oncogenes spontaneously develop T cell malignancies [9]. Here, we report the generation of a new model of inducible *BCL11B* knockout in *TCL1/LMO1* transgenic mice to study the role of this gene in T cell malignancies.

## 2. Results

Four transgenic mouse strains: *BCL11B^flox/WT^*, *Cre*^tg^, *SCL*^tg^ and *LMO1*^tg^ were sequentially crossed in order to obtain a *BCL11B^flox/flox^Cre*^tg/tg^*SCL*^tg^*LMO1*^tg^ mouse model of inducible *BCL11B* knockout in spontaneously developing T cell malignancies (Figure 1)**.**

### 2.1. Establishment of BCL11B^flox/flox^ LMO1^tg^ and BCL11B^flox/flox^ TAL1^tg^ Mice 

Heterozygous *BCL11B^flox/WT^* mice were crossed with either *LMO1*^tg^ or *TAL1*^tg^ transgenic mice (Figure 2A). Subsequently, *BCL11B^flox/WT^LMO1*^tg^ and *BCL11B^flox/WT^TAL1*^tg^ mice were crossed to generate homozygous *BCL11B^flox/flox^* mice with either the *LMO1* or *TAL1* oncogene: *BCL11B^flox/flox^LMO1*^tg^ and *BCL11B^flox/flox^TAL1*^tg^ (Figure 2B).

### 2.2. Introduction of the Cre/ESR1 Recombinase Construct to the BCL11B^flox/flox^LMO1^tg^ and BCL11B^flox/flox^TAL1^tg^ Mice

*BCL11B^flox/flox^LMO1*^tg^ and *BCL11B^flox/flox^TAL1*^tg^ mice were crossed with the heterozygous C57BL/6-*Gt(ROSA)26Sor^tm9(Cre/ESR1)Arte^* (*Cre^tg^*) mice. Subsequently, *BCL11B^flox/flox^LMO1*^tg^*Cre^tg^* and *BCL11B^flox/flox^TAL1*^tg^*Cre^tg^* were crossed with each other to obtain mice homozygous for the Cre construct: *BCL11B^flox/flox^LMO1*^tg^*Cre^tg/tg^* and *BCL11B^flox/flox^TAL1*^tg^*Cre^tg/tg^*. Those mice were healthy and reproduced normally. 

### 2.3. Generation of the Final Experimental Mouse Model of Inducible BCL11B Knockout in T-Cell Malignancies 

The *BCL11B^flox/flox^Cre*^tg/tg^*TAL1*^tg^ and *BCL11B^floxfloxl^CRE*^tg/tg^*LMO1*^tg^ animals were mated, and one month old progeny were genotyped via PCR from tail DNA. One fourth of the progeny carried both oncogenes (*BCL11B^flox/flox^LMO1^tg^/TAL1^tg^/Cre^tg/tg^*), and >90% of them developed T-ALL and were used for studies. The animals carrying only one oncogene, either *LMO1* (*BCL11B^flox/flox^CRE*^tg/tg^*LMO1*^tg^) or *TAL1* (*BCL11B^flox/flox^CRE*^tg/tg^*TAL1*^tg^), were healthy, bred normally and were used to maintain the mice in culture. Mice born without oncogenes (*BCL11B^flox/flox^CRE*^tg/tg^) were used as a control.

### 2.4. Creation of the BCL11B^flox/del^CRE^tg/tg^LMO1^tg^ and BCL11B^flox/del^CRE^tg/tg^TAL1^tg^ Mice

To increase the efficacy of the *BCL11B* knockout, a heterozygous mouse strain with an inherited deletion of the *BCL11B* gene (*BCL11B^flox/de^**^l^*) was generated by inducing gene deletion in one of the parents by tamoxifen administration. Surprisingly, almost all (29 out of 30) offspring were born with one deleted allele (Figure 3). For further breeding, *BCL11B^flox/del^* and *BCL11B^flox/flox^* mice were coupled to avoid the bi-allelic deletion of *BCL11B,* which was previously described [10] and also proven by us to be lethal. Similarly to the previously described animals with an intact *BCL11B* locus, the animals with *BCL11B^flox/del^CRE*^tg/tg^*LMO1*^tg^ mice were mated with the *BCL11B^flox/flox^Cre*^tg/tg^*TAL1*^tg^, or *BCL11B^flox/flox^CRE*^tg/tg^*LMO1*^tg^ with *BCL11B^flox/del^CRE*^tg/tg^*TAL1*^tg^. One month old progeny were genotyped via PCR from tail DNA. One fourth of the progeny carried both oncogenes, and half of them carried the *BCL11B* deletion (*BCL11B*^flox/del^*LMO1*^tg^/*TAL1*^tg^/Cre^tg/tg^).

### 2.5. Diagnosis of Hematologic Malignancies

As originally reported by Aplan et al., within 3–6 months after birth the progeny, carrying both oncogenes (*LMO1* and *TAL1*), spontaneously developed an aggressive T cell leukemia/lymphoma [9]. Starting from the fourth month of life, every two weeks the *LMO1*/*TAL1* mice were monitored for the development of malignancy. Lymphoma development was assessed by palpation and magnetic resonance imaging (MRI) (Figure 4), and the leukemia development was determined based on white blood cell (WBC) counting and peripheral blood smear analysis (Figure 5). Leukemia was diagnosed when the WBC was >10,000 cell/µL and/or the blast content >20%. 

### 2.6. Determination of the Immunophenotype of the Malignant Disease by Flow Cytometry

The diagnosis of T cell malignancy was further confirmed by FC of the peripheral blood. In contrast to healthy mice, in animals that developed T cell malignancies, besides an increased ratio of CD3+ cells, double-positive CD4+/CD8+, CD25+ or terminal deoxynucleotidyl transferase (TdT+) subsets of T cells were detected (Figure 6). The most common feature of all cases was the increased (30–90%) subset of CD3+ leukocytes (17/23; 74%). In 61% of the cases (14/23), double-positive CD4+/CD8+ T cells were observed, normally not present in the peripheral blood. In the majority of the CD4+/CD8+ cases, abnormal TdT+ (10/14; 71%) and CD25+ (8/14; 57%) cell subsets were also observed. In 10/13 (77%) of the analyzed cases, NK cell-specific antigen NK-1.1 was expressed, in half of the cases as a sole abnormal phenotype, in the other half accompanied by the CD4/CD8 phenotype, of which three also expressed TdT. 

## 3. Discussion

During the last two decades, extensive knowledge has been accumulated about the physiological role of *BCL11B* and the role of its mutations in inherited diseases. Although there are several reports on the involvement of *BCL11B* in hematological malignancies, including gene rearrangements, mutations and overexpression, the role of *BCL11B* in the development of the disease has not yet been conclusively clarified [11,12]. Additionally, the hypotheses on the possible mechanism of malignant transformation caused by *BCL11B* alterations are contradictory. While some studies postulated that *BCL11B* acted as a tumor suppressor and that inactivating mutations played a role in tumor development [13,14], others, including our studies, suggested *BCL11B* as being an oncogene [6,15] and its downregulation resulting in apoptosis of malignant T cells. To better understand the role of *BCL11B* in the pathogenesis of T cell malignancies and to test the efficacy of new therapeutic approaches, animal models of disease are needed.

In this study, we report the generation of a new model of inducible *BCL11B* knockout in *LMO1/TAL1* transgenic mice spontaneously developing T cell malignancies. In this model, the animals carrying only one oncogene, *LMO1*^tg^ or *TAL1*^tg^, are healthy and immunocompetent. They are easy to maintain in culture and breed normally. When crossed with each other, 90% of the double transgenic animals, *LMO1*^tg^/*TAL1*^tg^, within 3–6 months from birth, spontaneously develop T cell leukemia/lymphoma, resembling human T cell malignancies [9]. Since the animals possess the *BCL11B^flox/flox^CRE*^tg/tg^ genetic modification, the *BCL11B* gene can be inactivated at any time by induction of the Cre-lox knockout system, using synthetic estrogen (tamoxifen) administration.

To improve the efficacy of the *BCL11B* knockout, we modified that system and introduced an inherited knockout of one of the alleles. The animals were born with one floxed and one knocked-out *BCL11B* allele (*BCL11B^flox/del^CRE*^tg/tg^*LMO1*^tg^ and *BCL11B^flox/del^CRE*^tg/tg^*TAL1*^tg^), were healthy, bred normally, and passed the knocked-out allele to half of their offspring. As originally reported by Wakabayashi et al. [10], the germline knockout of both *BCL11B* alleles was lethal, and the animals died shortly after birth.

Besides the direct pro-apoptotic effect on malignant T cells, *BCL11B* suppression was shown to cause the transformation of normal T cells into induced T-to-natural killer (ITNK) cells [16]. Since ITNK cells killed tumor cells in vitro and in vivo, the inhibition of *BCL11B* may additionally exert its anti-tumor activity by inducing normal T cells to kill the remaining malignant T cells. In our previous studies, we showed that an increased expression of *BCL11B* leads to chemoresistance accompanied by G1 accumulation [17]. It can be speculated that the opposite, the suppression of *BCL11B*, should lead to an increased chemotherapy susceptibility. This suggests that combining *BCL11B* inhibition with chemotherapy could increase the efficacy of targeting cancer cells.

## 4. Materials and Methods

### 4.1. Mouse Models

Heterozygous transgenic mice bearing constructs which express TAL BHLH Transcription Factor 1 (*TAL1*;*SCL*) mRNA driven by a fusion to the STIL Centriolar Assembly Protein (*STIL*) exogenous promoter and LIM Domain Only 1 (*LMO1)*, mimicking common gene dysregulations associated with human T-ALL, were kindly provided by Peter Aplan (National Institute of Health/National Cancer Institute, Bethesda, MD, USA). These mice, bearing human oncogenes, *TAL1* or *LMO1*, responsible for T-cell acute lymphoblastic leukemia (T-ALL) development, are healthy and breed normally. When crossed with each other, the double transgenic mice *TAL1*^tg^/*LMO1*^tg^, within 3 to 6 months from birth, spontaneously develop T-cell leukemia/lymphoma [9].

The heterozygous *BCL11B* floxed mice were kindly provided by Philippe Kastner (Institut de Génétique et de Biologie Moléculaire et Cellulaire, Université de Strasbourg, France) with the permission of Mark Leid (Department of Integrative Biosciences, Oregon Health & Science University, Portland, OR, USA) [18].

The C57BL/6-*Gt(ROSA)26Sor^tm9(Cre/ESR1)Arte^* (*Cre^tg^*) mice, with inducible Cre recombinase, were purchased from Artemis (Koeln, Germany) [19]. Upon administration, synthetic estrogen (tamoxifen) binds to the estrogen receptor and activates the Cre recombinase. All studies involving animals were performed with the approval of the Local Ethical Committee for Animal Experiments in Poznan (approval code LKE 2/2018).

### 4.2. Genotyping

Genetic modifications were determined via PCR from tail DNA in one month old mice. For the determination of the presence of LoxP sequences within the *BCL11B* gene, the 5′ BCL6912f (mm39; chr12:107,883,573-592) TCGGAAGCCATGTGTGTTCT and the 3′ BCL7202r (mm39; chr12:107,883,863-844) TAGATCCCGTGTTCCCTTGC primers were used, amplifying a 291 bp fragment of the intron/fourth exon border of *BCL11B* in the wild type animals. In the *BCL11B* floxed mice, with the inserted LoxP sequence and fragments of the cloning vector, the amplicon was 342 bp long. 

For the presence of Rosa 26 Cre locus 1242_1: CCATCATCGAAGCTTCACTGAAG and 1242_2: GGAGTTTCAATACCCGAGATCATGC primers were used, amplifying a 315 bp fragment; and as a reaction control 1260_1: GAGACTCTGGCTACTCATCC and 1260_2: CCTTCAGCAAGAGCTGGGGAC primers were used, amplifying a 585 bp fragment of *CD79b*, as recommended by the supplier (Artemis). 

For the detection of the *STIL-TAL1* fusion product SILf: (hg38; chr1:47,314,077-058) GCTCCTACCCTGCAAACAGA and SCLr: (hg38; chr1:47,220,065-084) ATGTGTGGGGATCAGCTTGC primers were used, amplifying a 250 bp DNA fragment [9]. For the detection of *LMO1*, LMO1f: (chr11:8,230,382-363) AAGTGTGCGTGCTGTGACTG and LMO1r: (mm39; chr11:8,226,996-7,015) GCGAAGCAGTCGAGGTGATA primers were used, amplifying a 197 bp fragment of the spliced exons 2 and 3 [20].

### 4.3. Magnetic Resonance Imaging (MRI) 

MRI experiments were carried out using a preclinical horizontal scanner operating at 9.4 T (Agilent) equipped with a 600 mT/m gradient system, and a 40-mm i.d. quadrature millipede type coil was used. During the imaging experiment, animals were anesthetized with 2% isoflurane in a 20/80 air-oxygen mixture (induction 4% isoflurane) and put in a specially designed holder. The temperature of the animal was kept at 37 °C, and the respiration of the animal was also monitored and used to synchronize MRI experiments (1030, SA Instruments Inc., Stony Brook, NY, USA). 

MRI images of the proton spin density were collected at two localizations of each animal (abdomen area and neck area) using a fast spin-echo sequence (FSEMS) with the following parameters: TR = 5 s, effective TE = 10 ms, ETL = 8, FOV 35 × 35 mm, 2 averages, matrix size 256 × 256, slices from 12 to 30 depending on animal and localization (slice thickness 1 mm). 

### 4.4. Induction of BCL11B Knockout by Tamoxifen Administration

Tamoxifen (Sigma-Aldrich, Burlington, MA, USA) was dissolved in corn oil at a concentration of 10 mg/mL by shaking overnight at 37 °C. To induce Cre recombinase synthesis followed by knockout of the floxed *BCL11B* gene, the mice were injected intra-peritoneally with 1 mg of tamoxifen (100 µL of oil solution) for 3 subsequent days. The efficacy of deletion was checked 7 days after the third administration.

### 4.5. Detection of CreER-Mediated BCL11B Knockout

For the detection of the Cre-mediated deletion of the fourth exon of *BCL11B,* competitive PCR was performed with the BCL6912f, BCL7202r and BCL4812f (mm39; chr12:107,881,068-087) AATCCCACATGCCACTTTTC primers, amplifying, besides the floxed *BCL11B* locus (342 bp), the knocked-out locus (451 bp). 

### 4.6. Flow Cytometry

Flow cytometry analysis (FC) was performed on an Amnis^®^ FlowSight^®^ Flow Cytometer (Luminex; Austin, TX, USA) and CytoFLEX S Flow Cytometer (Beckman Coulter; Indianapolis, IN, USA). The following antibodies (BD Pharmigen, San Diego, CA, USA) were used for surface markers: Pacific Blue™ Hamster Anti-Mouse CD3e (558214), FITC Rat Anti-Mouse CD3 Molecular Complex (561798), PE Rat Anti-Mouse CD4 (553048), APC Rat Anti-Mouse CD8a (553035), APC Rat Anti-Mouse CD25 (557192), PE Rat Anti-Mouse CD44 (553134), PE Rat Anti-Mouse CD335 (NKp46) (560757), APC Mouse Anti-Mouse NK-1.1 (CD161) (550627). For the intracellular staining terminal-deoxynucleotidyl transferase (TdT): TdT Monoclonal Antibody (19-3), APC, eBioscience™ (17-5846-82) was used. Peripheral blood samples were treated with Erythrocyte Lysis Buffer (Qiagen, Germantown, MD, USA), and leukocytes were directly used for surface marker staining, while for intracellular staining, cells were fixed and permeabilized using BD Cytofix/Cytoperm Kit (554714) and BD Cytoperm™ Permeabilization Buffer Plus (561651). The cut-off for positivity was set at 10% for CD4/CD8, CD25 and TdT, and at 20% for NK1.1. antibodies. 

## 5. Conclusions

The mouse model we created can be used to study the function of *BCL11B* in T cell malignancies, especially the effect of *BCL11B* suppression. Since, to date, no specific *BCL11B* inhibitor has been invented, this model can be used as a proof of principle for the rationale of therapeutic *BCL11B* suppression in T cell neoplasms. 

## Figures and Tables

**Figure 1 ijms-23-04932-f001:**
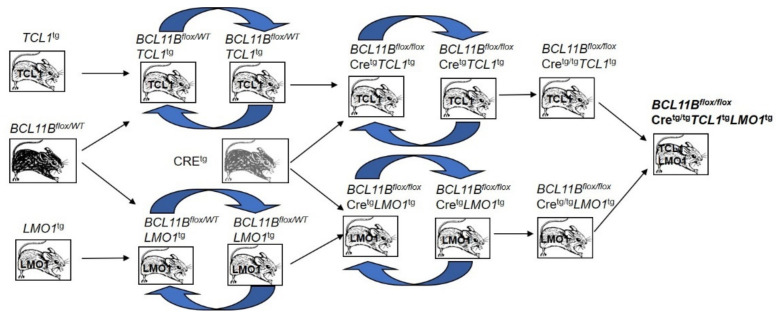
Scheme of transgenic mice crossing to generate inducible *BCL11B* knockout in spontaneously developing T cell malignancies.

**Figure 2 ijms-23-04932-f002:**
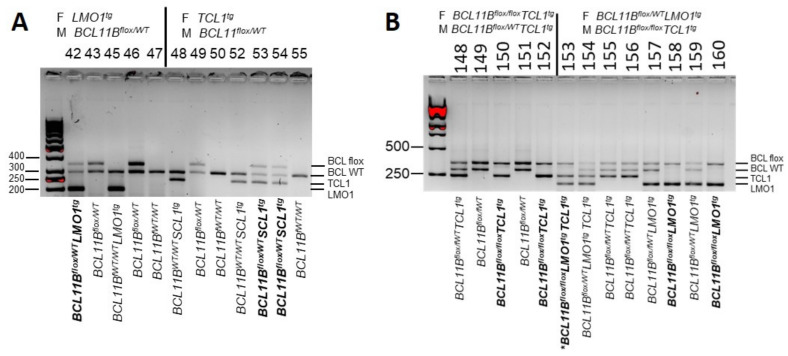
Establishment of *BCL11B^flox/flox^ LMO1*^tg^ and *BCL11B^flox/flox^ TAL1*^tg^ mice. (**A**) Generation of heterozygous *BCL11B^flox/WT^LMO1*^tg^ and *BCL11B^flox/WT^TAL1*^tg^ mice. Genotypes of the parents (F: female, M: male) are indicated on the top, the identification numbers of the progeny below, the photo the genotypes of the progeny below, the size of the DNA ladder in bp on the left, and the location of multiplex PCR products on the right. The *BCL11B^flox/WT^LMO1*^tg^ and *BCL11B^flox/WT^TAL1*^tg^ mice used for further crossing are in bold. MW 1: Molecular weight marker 1—GeneRuler 50 bp (Thermo Scientific, Waltham, MA, USA). (**B**) Generation of homozygous *BCL11B^flox/fox^LMO1*^tg^ and *BCL11B^flox/flox^TAL1*^tg^ mice. The homozygous *BCL11B^flox/flox^* mice with either *LMO1*^tg^ or *TAL1*^tg^ oncogenes are in bold. The homozygous *BCL11B^flox/flox^* mouse with both oncogenes is in bold and is indicated by an asterisk (*). MW 2: Molecular weight marker 2—GeneRuler 1 kb (Thermo Scientific).

**Figure 3 ijms-23-04932-f003:**
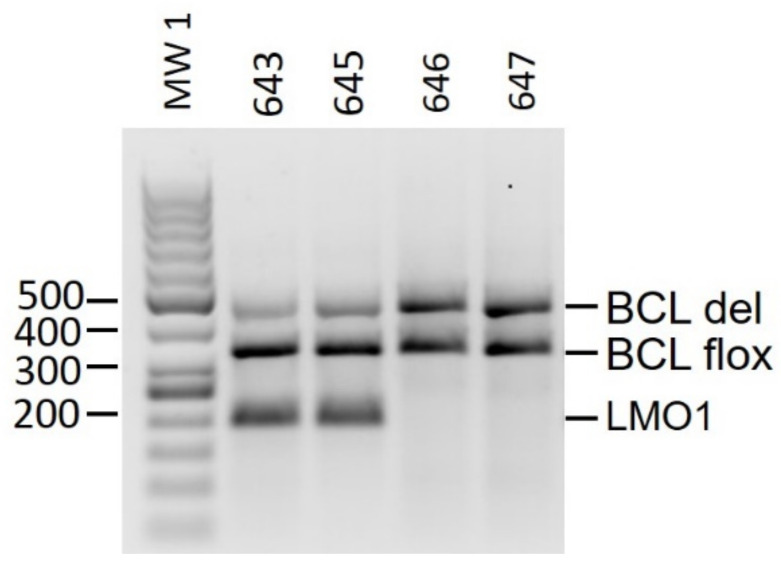
Creation of the *BCL11B^flox/del^CRE*^tg/tg^*TAL1*^tg^ mice with germline hemizygous *BCL11B* knockout. Mice 643 and 645 are offspring of the tamoxifen-induced *BCL11B* knockout mother with *LMO1*^tg^ and untreated father. Mice 646 and 647 are offspring of the tamoxifen-induced *BCL11B* knockout mother and untreated father, both without oncogenes. MW 1: Molecular weight marker 1—GeneRuler 50 bp (Thermo Scientific).

**Figure 4 ijms-23-04932-f004:**
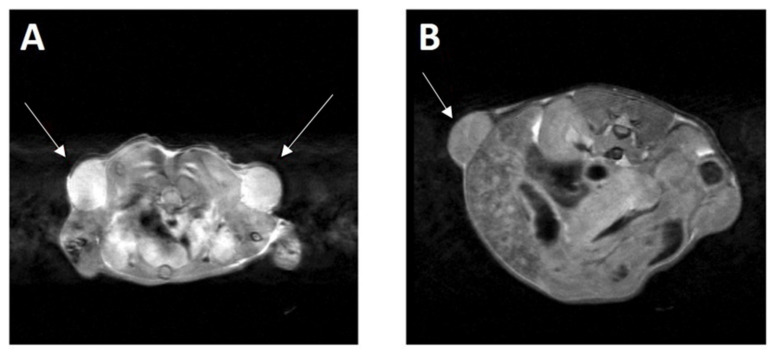
Detection of tumors using magnetic resonance imaging. (**A**) Bilateral neck lymph node tumors, (**B**) abdominal lymph node tumor.

**Figure 5 ijms-23-04932-f005:**
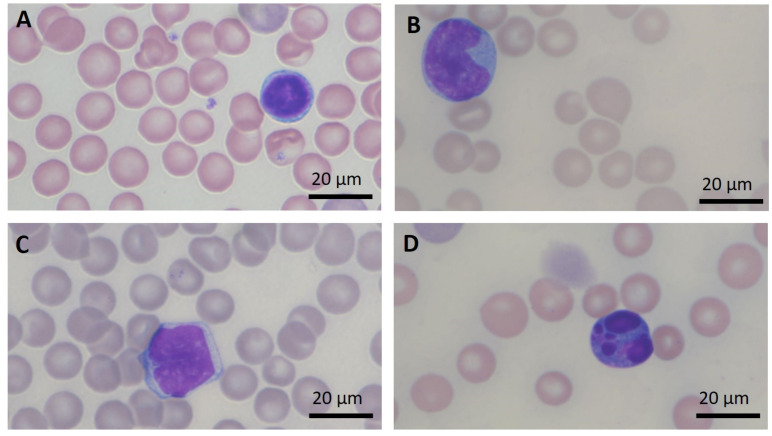
Peripheral blood smear at diagnosis. Magnification 1000×. (**A**) Normal lymphocyte, (**B**,**C**) lymphoblasts, (**D**) apoptotic cell.

**Figure 6 ijms-23-04932-f006:**
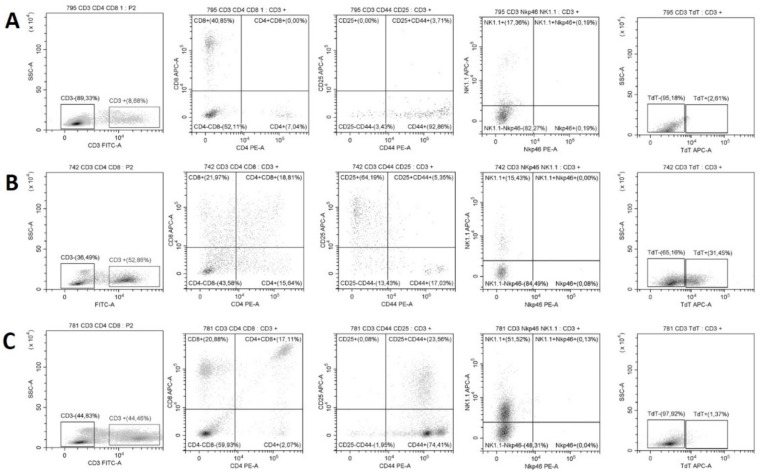
Determination of malignant T cell immunophenotype by flow cytometry analysis. (**A**) Mouse 795 without signs of malignancy, (**B**) Mouse 742 with increased ratio of CD3+, CD4+/CD8+, CD25+/CD44-, and TdT+ cells. (**C**) Mouse 781 with increased ratio of CD3+, CD4+/CD8+, CD25+/CD44+ and NK-1.1 cells.

## Data Availability

Not applicable.

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
