# Peer review of "Generation of Inducible BCL11B Knockout in TAL1/LMO1 Transgenic Mouse T Cell Leukemia/Lymphoma Model"

_ijms, 2022, doi:10.3390/ijms23094932_

Round 1
Reviewer 1 Report
Authors investigated the role of BCL11B in T cell malignancy. They generated the BCL11B knockout in SCL/LMO1 transgenic mouse which is the model of T cell leukemia/lymphoma.
They described SCL or LMO1 single transgenic models that were not able to generate T cell malignancies. However, the double transgenic model yielded the T cell malignancies.
Authors mentioned in this double transgenic model, they could knockout BCL11B which is also designed in these transgenic models and is able to be deleted by tamoxifen and CRE-activation.
However, cellular and molecular details of comparison between positive or negative BCL11B in SCL/LMO1 double transgenic models were not oriented.
Thus it would be better to know what happens in the knockout BCL11BN model which compares non-knockouted BCL11B models in SCL/LMO1 double transgenic models.
Overall, we have to say this investigation is still preliminary.
In addition, in the introduction, the details of SCL and LMO1 would be described in detail. Authors' descriptions are too short and summarized. THe similar tendency is also seen in Discussion part.
Author Response
However, cellular and molecular details of comparison between positive or negative BCL11B in SCL/LMO1 double transgenic models were not oriented. Thus it would be better to know what happens in the knockout BCL11BN model which compares non-knockouted BCL11B models in SCL/LMO1 double transgenic models.
Overall, we have to say this investigation is still preliminary.
In our paper we report just the generation of a new mouse model. We absolutely agree that it would be better to know what happens upon the knockout. We are currently running experiments to unravel the biological and potential effect of the BCL11B knockout. Since the animal experiments take a long time, we will not be able to obtain the results before the paper submission deadline.
In addition, in the introduction, the details of SCL and LMO1 would be described in detail. Authors' descriptions are too short and summarized. THe similar tendency is also seen in Discussion part.
We have now included the information on SCL and LMO1 genes in the Introduction, although those genes are only triggers of the malignancy, and are not the subject of this study. We have also added to the Discussion new potential mode of action of BCL11B inhibitors.
Reviewer 2 Report
The authors present an interesting and innovative methodological paper in which they report the generation of a new model of an inducible BCL11B knockout in transgenic mice which are prone to develop T-cell malignancies.
Specific Points of Criticism and Suggestions for Alterations:
- The official designation of the gene SCL/TAL1 is "TAL1" according to Human Genome Organisation (HUGO) Gene Nomenclature Committee: https://www.genenames.org /
Specifically for TAL1 see: https://www.genenames.org/data/gene-symbol-report/#!/hgnc_id/HGNC:11556 - "SCL" is only an alias symbol.
Hence throughout the manuscript including the title "TAL1" should be used instead of "SCL".
LMO1 is indeed the approved symbol: https://www.genenames.org/data/gene-symbol-report/#!/hgnc_id/HGNC:6641
BCL11B is indeed the approved symbol: https://www.genenames.org/data/gene-symbol-report/#!/hgnc_id/HGNC:13222
STIL is indeed the approved symbol: https://www.genenames.org/data/gene-symbol-report/#!/hgnc_id/HGNC:10879
- Lines 30-32: unusual English, sentence should be reworded.
- Lines 186-187: „TSIL“ or „STIL“?
- 4.6. Flow Cytometry: What was the cut-off percentage for positivity and negativity for a given marker? To be indicated in this section.
- Line 253: Does the antibody NK-1.1 detect the CD161? if so, it might be indicated on line 253.
Author Response
- The official designation of the gene SCL/TAL1 is "TAL1" according to Human Genome Organisation (HUGO) Gene Nomenclature Committee: https://www.genenames.org /
Specifically for TAL1 see: https://www.genenames.org/data/gene-symbol-report/#!/hgnc_id/HGNC:11556 - "SCL" is only an alias symbol.
Hence throughout the manuscript including the title "TAL1" should be used instead of "SCL".
We have now replaced the SCL by TAL1 throughout the document, including the title and the figures.
- Lines 30-32: unusual English, sentence should be reworded.
The sentence has now been corrected to “The mouse model of inducible BCL11B knockout we generated can be used to study its role in cancer development and the potential therapeutic effect of BCL11B inhibition in T-cell leukemia and lymphoma.”
- Lines 186-187: „TSIL“ or „STIL“?
It should read STIL. The typing error has now been corrected.
- 4.6. Flow Cytometry: What was the cut-off percentage for positivity and negativity for a given marker? To be indicated in this section.
The information „The cut-off for positivity was set at 10% for CD4/CD8, CD25 and TdT, and at 20% for NK1.1. antibodies.” Has now been included in the 4.6 section.
- Line 253: Does the antibody NK-1.1 detect the CD161? if so, it might be indicated on line 253.
Yes, the antibody NK-1.1 detects the CD161 antigen. This information has now been indicated.
Round 2
Reviewer 1 Report
In this revision, authors added the explanation regarding SCL and LMO1. In addition, some description was added in the discussion part. However, even authors wrote as "we will not be able to obtain the results before the paper submission deadline", just the generation only did not explain and reveal any roles of BCL11B in T cell malignancies. Authors could wait to submit manuscript after they get enough and nice findings regardless of the deadline for this special issue.Author Response
Thank you for accepting the additions we have made in the Introduction and Discussion sections. As we have explained in the first revision: the requested animal experiments are long-lasting and we will not be able to obtain the results before the paper submission deadline of this special issue.